# Teaching Machines to Read and Comprehend

**Karl Moritz Hermann**[†]    **Tomáš Kočiský**[†‡]    **Edward Grefenstette**[†]
**Lasse Espeholt**[†]    **Will Kay**[†]    **Mustafa Suleyman**[†]    **Phil Blunsom**[†‡]
[†]Google DeepMind    [‡]University of Oxford
{kmh,tkocisky,etg,lespeholt,wkay,mustafasul,pblunsom}@google.com

## Abstract

Teaching machines to read natural language documents remains an elusive challenge. Machine reading systems can be tested on their ability to answer questions posed on the contents of documents that they have seen, but until now large scale training and test datasets have been missing for this type of evaluation. In this work we define a new methodology that resolves this bottleneck and provides large scale supervised reading comprehension data. This allows us to develop a class of attention based deep neural networks that learn to read real documents and answer complex questions with minimal prior knowledge of language structure.

## 1 Introduction

Progress on the path from shallow bag-of-words information retrieval algorithms to machines capable of reading and understanding documents has been slow. Traditional approaches to machine reading and comprehension have been based on either hand engineered grammars [1], or information extraction methods of detecting predicate argument triples that can later be queried as a relational database [2]. Supervised machine learning approaches have largely been absent from this space due to both the lack of large scale training datasets, and the difficulty in structuring statistical models flexible enough to learn to exploit document structure.

While obtaining supervised natural language reading comprehension data has proved difficult, some researchers have explored generating synthetic narratives and queries [3, 4]. Such approaches allow the generation of almost unlimited amounts of supervised data and enable researchers to isolate the performance of their algorithms on individual simulated phenomena. Work on such data has shown that neural network based models hold promise for modelling reading comprehension, something that we will build upon here. Historically, however, many similar approaches in Computational Linguistics have failed to manage the transition from synthetic data to real environments, as such closed worlds inevitably fail to capture the complexity, richness, and noise of natural language [5].

In this work we seek to directly address the lack of real natural language training data by introducing a novel approach to building a supervised reading comprehension data set. We observe that summary and paraphrase sentences, with their associated documents, can be readily converted to context–query–answer triples using simple entity detection and anonymisation algorithms. Using this approach we have collected two new corpora of roughly a million news stories with associated queries from the CNN and Daily Mail websites.

We demonstrate the efficacy of our new corpora by building novel deep learning models for reading comprehension. These models draw on recent developments for incorporating attention mechanisms into recurrent neural network architectures [6, 7, 8, 4]. This allows a model to focus on the aspects of a document that it believes will help it answer a question, and also allows us to visualises its inference process. We compare these neural models to a range of baselines and heuristic benchmarks based upon a traditional frame semantic analysis provided by a state-of-the-art natural language processing

| | CNN | | | Daily Mail | | |
|---|---|---|---|---|---|---|
| | train | valid | test | train | valid | test |
| # months | 95 | 1 | 1 | 56 | 1 | 1 |
| # documents | 90,266 | 1,220 | 1,093 | 196,961 | 12,148 | 10,397 |
| # queries | 380,298 | 3,924 | 3,198 | 879,450 | 64,835 | 53,182 |
| Max # entities | 527 | 187 | 396 | 371 | 232 | 245 |
| Avg # entities | 26.4 | 26.5 | 24.5 | 26.5 | 25.5 | 26.0 |
| Avg # tokens | 762 | 763 | 716 | 813 | 774 | 780 |
| Vocab size | 118,497 | | | 208,045 | | |

Table 1: Corpus statistics. Articles were collected starting in April 2007 for CNN and June 2010 for the Daily Mail, both until the end of April 2015. Validation data is from March, test data from April 2015. Articles of over 2000 tokens and queries whose answer entity did not appear in the context were filtered out.

| Top N | Cumulative % | |
|---|---|---|
| | CNN | Daily Mail |
| 1 | 30.5 | 25.6 |
| 2 | 47.7 | 42.4 |
| 3 | 58.1 | 53.7 |
| 5 | 70.6 | 68.1 |
| 10 | 85.1 | 85.5 |

Table 2: Percentage of time that the correct answer is contained in the top $N$ most frequent entities in a given document.

(NLP) pipeline. Our results indicate that the neural models achieve a higher accuracy, and do so without any specific encoding of the document or query structure.

## 2 Supervised training data for reading comprehension

The reading comprehension task naturally lends itself to a formulation as a supervised learning problem. Specifically we seek to estimate the conditional probability $p(a|c,q)$, where $c$ is a context document, $q$ a query relating to that document, and $a$ the answer to that query. For a focused evaluation we wish to be able to exclude additional information, such as world knowledge gained from co-occurrence statistics, in order to test a model's core capability to detect and understand the linguistic relationships between entities in the context document.

Such an approach requires a large training corpus of document–query–answer triples and until now such corpora have been limited to hundreds of examples and thus mostly of use only for testing [9]. This limitation has meant that most work in this area has taken the form of unsupervised approaches which use templates or syntactic/semantic analysers to extract relation tuples from the document to form a knowledge graph that can be queried.

Here we propose a methodology for creating real-world, large scale supervised training data for learning reading comprehension models. Inspired by work in summarisation [10, 11], we create two machine reading corpora by exploiting online newspaper articles and their matching summaries. We have collected 93k articles from the CNN[1] and 220k articles from the Daily Mail[2] websites. Both news providers supplement their articles with a number of bullet points, summarising aspects of the information contained in the article. Of key importance is that these summary points are abstractive and do not simply copy sentences from the documents. We construct a corpus of document–query–answer triples by turning these bullet points into Cloze [12] style questions by replacing one entity at a time with a placeholder. This results in a combined corpus of roughly 1M data points (Table 1). Code to replicate our datasets—and to apply this method to other sources—is available online[3].

### 2.1 Entity replacement and permutation

Note that the focus of this paper is to provide a corpus for evaluating a model's ability to read and comprehend a single document, not world knowledge or co-occurrence. To understand that distinction consider for instance the following Cloze form queries (created from headlines in the Daily Mail validation set): *a*) The hi-tech bra that helps you beat breast **X**; *b*) Could Saccharin help beat **X** ?; *c*) Can fish oils help fight prostate **X** ? An ngram language model trained on the Daily Mail would easily correctly predict that (**X** = *cancer*), regardless of the contents of the context document, simply because this is a very frequently cured entity in the Daily Mail corpus.

| Original Version | Anonymised Version |
|---|---|
| **Context** | |
| The BBC producer allegedly struck by Jeremy Clarkson will not press charges against the "Top Gear" host, his lawyer said Friday. Clarkson, who hosted one of the most-watched television shows in the world, was dropped by the BBC Wednesday after an internal investigation by the British broadcaster found he had subjected producer Oisin Tymon "to an unprovoked physical and verbal attack." ... | the *ent381* producer allegedly struck by *ent212* will not press charges against the " *ent153* " host , his lawyer said friday . *ent212* , who hosted one of the most - watched television shows in the world , was dropped by the *ent381* wednesday after an internal investigation by the *ent180* broadcaster found he had subjected producer *ent193* " to an unprovoked physical and verbal attack . " ... |
| **Query** | |
| Producer **X** will not press charges against Jeremy Clarkson, his lawyer says. | producer **X** will not press charges against *ent212* , his lawyer says . |
| **Answer** | |
| Oisin Tymon | *ent193* |

Table 3: Original and anonymised version of a data point from the Daily Mail validation set. The anonymised entity markers are constantly permuted during training and testing.

To prevent such degenerate solutions and create a focused task we anonymise and randomise our corpora with the following procedure, *a*) use a coreference system to establish coreferents in each data point; *b*) replace all entities with abstract entity markers according to coreference; *c*) randomly permute these entity markers whenever a data point is loaded.

Compare the original and anonymised version of the example in Table 3. Clearly a human reader can answer both queries correctly. However in the anonymised setup the context document is required for answering the query, whereas the original version could also be answered by someone with the requisite background knowledge. Therefore, following this procedure, the only remaining strategy for answering questions is to do so by exploiting the context presented with each question. Thus performance on our two corpora truly measures reading comprehension capability. Naturally a production system would benefit from using all available information sources, such as clues through language and co-occurrence statistics.

Table 2 gives an indication of the difficulty of the task, showing how frequent the correct answer is contained in the top $N$ entity markers in a given document. Note that our models don't distinguish between entity markers and regular words. This makes the task harder and the models more general.

# 3 Models

So far we have motivated the need for better datasets and tasks to evaluate the capabilities of machine reading models. We proceed by describing a number of baselines, benchmarks and new models to evaluate against this paradigm. We define two simple baselines, the majority baseline (`maximum frequency`) picks the entity most frequently observed in the context document, whereas the exclusive majority (`exclusive frequency`) chooses the entity most frequently observed in the context but not observed in the query. The idea behind this exclusion is that the placeholder is unlikely to be mentioned twice in a single Cloze form query.

## 3.1 Symbolic Matching Models

Traditionally, a pipeline of NLP models has been used for attempting question answering, that is models that make heavy use of linguistic annotation, structured world knowledge and semantic parsing and similar NLP pipeline outputs. Building on these approaches, we define a number of NLP-centric models for our machine reading task.

**Frame-Semantic Parsing** Frame-semantic parsing attempts to identify predicates and their arguments, allowing models access to information about "who did what to whom". Naturally this kind of annotation lends itself to being exploited for question answering. We develop a benchmark that

makes use of frame-semantic annotations which we obtained by parsing our model with a state-of-the-art frame-semantic parser [13, 14]. As the parser makes extensive use of linguistic information we run these benchmarks on the unanonymised version of our corpora. There is no significant advantage in this as the frame-semantic approach used here does not possess the capability to generalise through a language model beyond exploiting one during the parsing phase. Thus, the key objective of evaluating machine comprehension abilities is maintained. Extracting entity-predicate triples—denoted as $(e_1, V, e_2)$—from both the query $q$ and context document $d$, we attempt to resolve queries using a number of rules with an increasing recall/precision trade-off as follows (Table 4).

|   | Strategy | Pattern $\in q$ | Pattern $\in d$ | Example (Cloze / Context) |
|---|---|---|---|---|
| 1 | Exact match | $(p, V, y)$ | $(\boldsymbol{x}, V, y)$ | X loves Suse / **Kim** loves Suse |
| 2 | be.01.V match | $(p, be.01.V, y)$ | $(\boldsymbol{x}, be.01.V, y)$ | X is president / **Mike** is president |
| 3 | Correct frame | $(p, V, y)$ | $(\boldsymbol{x}, V, z)$ | X won Oscar / **Tom** won Academy Award |
| 4 | Permuted frame | $(p, V, y)$ | $(y, V, \boldsymbol{x})$ | X met Suse / Suse met **Tom** |
| 5 | Matching entity | $(p, V, y)$ | $(\boldsymbol{x}, Z, y)$ | X likes candy / **Tom** loves candy |
| 6 | Back-off strategy | *Pick the most frequent entity from the context that doesn't appear in the query* | | |

Table 4: Resolution strategies using PropBank triples. $\boldsymbol{x}$ denotes the entity proposed as answer, $V$ is a fully qualified PropBank frame (e.g. *give.01.V*). Strategies are ordered by precedence and answers determined accordingly. This heuristic algorithm was iteratively tuned on the validation data set.

For reasons of clarity, we pretend that all PropBank triples are of the form $(e_1, V, e_2)$. In practice, we take the argument numberings of the parser into account and only compare like with like, except in cases such as the permuted frame rule, where ordering is relaxed. In the case of multiple possible answers from a single rule, we randomly choose one.

**Word Distance Benchmark**   We consider another baseline that relies on word distance measurements. Here, we align the placeholder of the Cloze form question with each possible entity in the context document and calculate a distance measure between the question and the context around the aligned entity. This score is calculated by summing the distances of every word in $q$ to their nearest aligned word in $d$, where alignment is defined by matching words either directly or as aligned by the coreference system. We tune the maximum penalty per word ($m = 8$) on the validation data.

### 3.2   Neural Network Models

Neural networks have successfully been applied to a range of tasks in NLP. This includes classification tasks such as sentiment analysis [15] or POS tagging [16], as well as generative problems such as language modelling or machine translation [17]. We propose three neural models for estimating the probability of word type $a$ from document $d$ answering query $q$:

$$p(a|d, q) \propto \exp\left(W(a)g(d, q)\right), \quad \text{s.t. } a \in V,$$

where $V$ is the vocabulary[4], and $W(a)$ indexes row $a$ of weight matrix $W$ and through a slight abuse of notation word types double as indexes. Note that we do not privilege entities or variables, the model must learn to differentiate these in the input sequence. The function $g(d, q)$ returns a vector embedding of a document and query pair.

**The Deep LSTM Reader**   Long short-term memory (LSTM, [18]) networks have recently seen considerable success in tasks such as machine translation and language modelling [17]. When used for translation, Deep LSTMs [19] have shown a remarkable ability to embed long sequences into a vector representation which contains enough information to generate a full translation in another language. Our first neural model for reading comprehension tests the ability of Deep LSTM encoders to handle significantly longer sequences. We feed our documents one word at a time into a Deep LSTM encoder, after a delimiter we then also feed the query into the encoder. Alternatively we also experiment with processing the query then the document. The result is that this model processes each document query pair as a single long sequence. Given the embedded document and query the network predicts which token in the document answers the query.

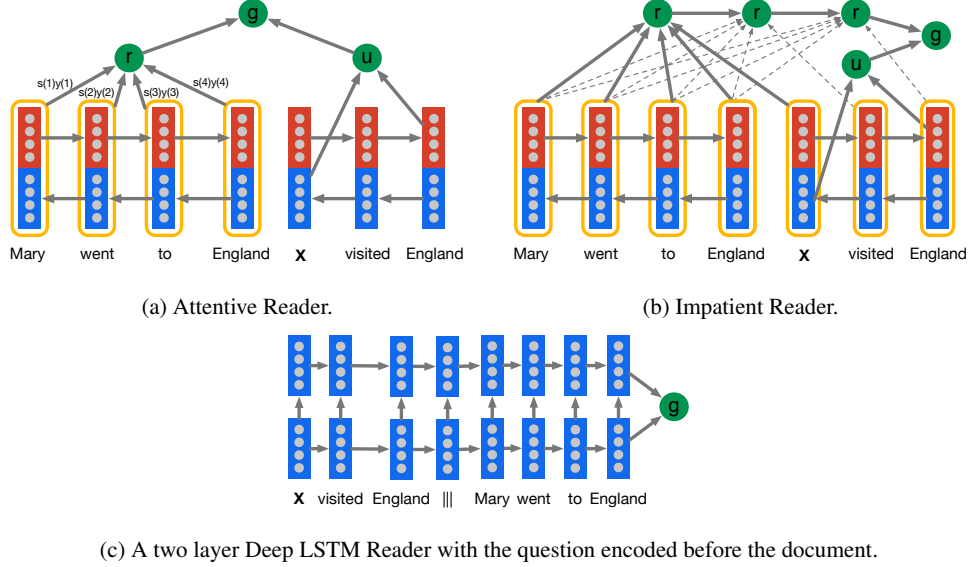

(a) Attentive Reader.　　　　　　　　(b) Impatient Reader.

(c) A two layer Deep LSTM Reader with the question encoded before the document.

Figure 1: Document and query embedding models.

We employ a Deep LSTM cell with skip connections from each input $x(t)$ to every hidden layer, and from every hidden layer to the output $y(t)$:

$$x'(t,k) = x(t)||y'(t,k-1), \qquad y(t) = y'(t,1)||\dots||y'(t,K)$$
$$i(t,k) = \sigma\left(W_{kxi}x'(t,k) + W_{khi}h(t-1,k) + W_{kci}c(t-1,k) + b_{ki}\right)$$
$$f(t,k) = \sigma\left(W_{kxf}x(t) + W_{khf}h(t-1,k) + W_{kcf}c(t-1,k) + b_{kf}\right)$$
$$c(t,k) = f(t,k)c(t-1,k) + i(t,k)\tanh\left(W_{kxc}x'(t,k) + W_{khc}h(t-1,k) + b_{kc}\right)$$
$$o(t,k) = \sigma\left(W_{kxo}x'(t,k) + W_{kho}h(t-1,k) + W_{kco}c(t,k) + b_{ko}\right)$$
$$h(t,k) = o(t,k)\tanh\left(c(t,k)\right)$$
$$y'(t,k) = W_{ky}h(t,k) + b_{ky}$$

where $||$ indicates vector concatenation $h(t,k)$ is the hidden state for layer $k$ at time $t$, and $i$, $f$, $o$ are the input, forget, and output gates respectively. Thus our Deep LSTM Reader is defined by $g^{\text{LSTM}}(d,q) = y(|d| + |q|)$ with input $x(t)$ the concatenation of $d$ and $q$ separated by the delimiter $|||$.

**The Attentive Reader**　　The Deep LSTM Reader must propagate dependencies over long distances in order to connect queries to their answers. The fixed width hidden vector forms a bottleneck for this information flow that we propose to circumvent using an attention mechanism inspired by recent results in translation and image recognition [6, 7]. This attention model first encodes the document and the query using separate bidirectional single layer LSTMs [19].

We denote the outputs of the forward and backward LSTMs as $\overrightarrow{y}(t)$ and $\overleftarrow{y}(t)$ respectively. The encoding $u$ of a query of length $|q|$ is formed by the concatenation of the final forward and backward outputs, $u = \overrightarrow{y_q}(|q|) \,||\, \overleftarrow{y_q}(1)$.

For the document the composite output for each token at position $t$ is, $y_d(t) = \overrightarrow{y_d}(t) \,||\, \overleftarrow{y_d}(t)$. The representation $r$ of the document $d$ is formed by a weighted sum of these output vectors. These weights are interpreted as the degree to which the network attends to a particular token in the document when answering the query:

$$m(t) = \tanh\left(W_{ym}y_d(t) + W_{um}u\right),$$
$$s(t) \propto \exp\left(\mathsf{w}_{ms}^{\mathsf{T}}m(t)\right),$$
$$r = y_d s,$$

where we are interpreting $y_d$ as a matrix with each column being the composite representation $y_d(t)$ of document token $t$. The variable $s(t)$ is the normalised attention at token $t$. Given this attention

score the embedding of the document $r$ is computed as the weighted sum of the token embeddings. The model is completed with the definition of the joint document and query embedding via a non-linear combination:

$$g^{\text{AR}}(d, q) = \tanh\left(W_{rg}r + W_{ug}u\right).$$

The Attentive Reader can be viewed as a generalisation of the application of Memory Networks to question answering [3]. That model employs an attention mechanism at the sentence level where each sentence is represented by a bag of embeddings. The Attentive Reader employs a finer grained token level attention mechanism where the tokens are embedded given their entire future and past context in the input document.

**The Impatient Reader**   The Attentive Reader is able to focus on the passages of a context document that are most likely to inform the answer to the query. We can go further by equipping the model with the ability to reread from the document as each query token is read. At each token $i$ of the query $q$ the model computes a document representation vector $r(i)$ using the bidirectional embedding $y_q(i) = \overrightarrow{y_q}(i) \;||\; \overleftarrow{y_q}(i)$:

$$m(i, t) = \tanh\left(W_{dm}y_d(t) + W_{rm}r(i-1) + W_{qm}y_q(i)\right), \quad 1 \leq i \leq |q|,$$
$$s(i, t) \propto \exp\left(\mathrm{w}_{ms}^{\mathsf{T}}m(i, t)\right),$$
$$r(0) = \mathbf{r_0}, \quad r(i) = y_d^{\mathsf{T}}s(i) + \tanh\left(W_{rr}r(i-1)\right) \quad 1 \leq i \leq |q|.$$

The result is an attention mechanism that allows the model to recurrently accumulate information from the document as it sees each query token, ultimately outputting a final joint document query representation for the answer prediction,

$$g^{\text{IR}}(d, q) = \tanh\left(W_{rg}r(|q|) + W_{qg}u\right).$$

## 4   Empirical Evaluation

Having described a number of models in the previous section, we next evaluate these models on our reading comprehension corpora. Our hypothesis is that neural models should in principle be well suited for this task. However, we argued that simple recurrent models such as the LSTM probably have insufficient expressive power for solving tasks that require complex inference. We expect that the attention-based models would therefore outperform the pure LSTM-based approaches.

Considering the second dimension of our investigation, the comparison of traditional versus neural approaches to NLP, we do not have a strong prior favouring one approach over the other. While numerous publications in the past few years have demonstrated neural models outperforming classical methods, it remains unclear how much of that is a side-effect of the language modelling capabilities intrinsic to any neural model for NLP. The entity anonymisation and permutation aspect of the task presented here may end up levelling the playing field in that regard, favouring models capable of dealing with syntax rather than just semantics.

With these considerations in mind, the experimental part of this paper is designed with a three-fold aim. First, we want to establish the difficulty of our machine reading task by applying a wide range of models to it. Second, we compare the performance of parse-based methods versus that of neural models. Third, within the group of neural models examined, we want to determine what each component contributes to the end performance; that is, we want to analyse the extent to which an LSTM can solve this task, and to what extent various attention mechanisms impact performance.

All model hyperparameters were tuned on the respective validation sets of the two corpora.[5] Our experimental results are in Table 5, with the Attentive and Impatient Readers performing best across both datasets.

| | CNN | | Daily Mail | |
|---|---|---|---|---|
| | valid | test | valid | test |
| Maximum frequency | 30.5 | 33.2 | 25.6 | 25.5 |
| Exclusive frequency | 36.6 | 39.3 | 32.7 | 32.8 |
| Frame-semantic model | 36.3 | 40.2 | 35.5 | 35.5 |
| Word distance model | 50.5 | 50.9 | 56.4 | 55.5 |
| Deep LSTM Reader | 55.0 | 57.0 | 63.3 | 62.2 |
| Uniform Reader | 39.0 | 39.4 | 34.6 | 34.4 |
| Attentive Reader | 61.6 | 63.0 | **70.5** | **69.0** |
| Impatient Reader | **61.8** | **63.8** | 69.0 | 68.0 |

Table 5: Accuracy of all the models and benchmarks on the CNN and Daily Mail datasets. The Uniform Reader baseline sets all of the $m(t)$ parameters to be equal.

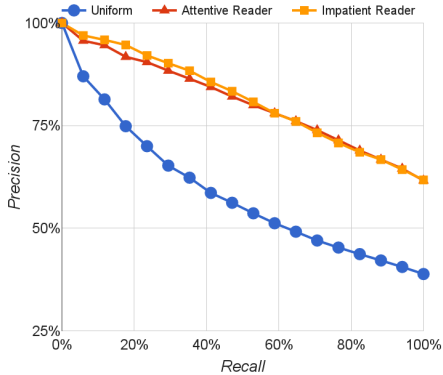

Figure 2: Precision@Recall for the attention models on the CNN validation data.

**Frame-semantic benchmark**   While the one frame-semantic model proposed in this paper is clearly a simplification of what could be achieved with annotations from an NLP pipeline, it does highlight the difficulty of the task when approached from a symbolic NLP perspective.

Two issues stand out when analysing the results in detail. First, the frame-semantic pipeline has a poor degree of coverage with many relations not being picked up by our PropBank parser as they do not adhere to the default predicate-argument structure. This effect is exacerbated by the type of language used in the highlights that form the basis of our datasets. The second issue is that the frame-semantic approach does not trivially scale to situations where several sentences, and thus frames, are required to answer a query. This was true for the majority of queries in the dataset.

**Word distance benchmark**   More surprising perhaps is the relatively strong performance of the word distance benchmark, particularly relative to the frame-semantic benchmark, which we had expected to perform better. Here, again, the nature of the datasets used can explain aspects of this result. Where the frame-semantic model suffered due to the language used in the highlights, the word distance model benefited. Particularly in the case of the Daily Mail dataset, highlights frequently have significant lexical overlap with passages in the accompanying article, which makes it easy for the word distance benchmark. For instance the query "*Tom Hanks is friends with* **X***'s manager, Scooter Brown*" has the phrase "*... turns out he is good friends with Scooter Brown, manager for Carly Rae Jepson*" in the context. The word distance benchmark correctly aligns these two while the frame-semantic approach fails to pickup the friendship or management relations when parsing the query. We expect that on other types of machine reading data where questions rather than Cloze queries are used this particular model would perform significantly worse.

**Neural models**   Within the group of neural models explored here, the results paint a clear picture with the Impatient and the Attentive Readers outperforming all other models. This is consistent with our hypothesis that attention is a key ingredient for machine reading and question answering due to the need to propagate information over long distances. The Deep LSTM Reader performs surprisingly well, once again demonstrating that this simple sequential architecture can do a reasonable job of learning to abstract long sequences, even when they are up to two thousand tokens in length. However this model does fail to match the performance of the attention based models, even though these only use single layer LSTMs.[6]

The poor results of the Uniform Reader support our hypothesis of the significance of the attention mechanism in the Attentive model's performance as the only difference between these models is that the attention variables are ignored in the Uniform Reader. The precision@recall statistics in Figure 2 again highlight the strength of the attentive approach.

We can visualise the attention mechanism as a heatmap over a context document to gain further insight into the models' performance. The highlighted words show which tokens in the document were attended to by the model. In addition we must also take into account that the vectors at each

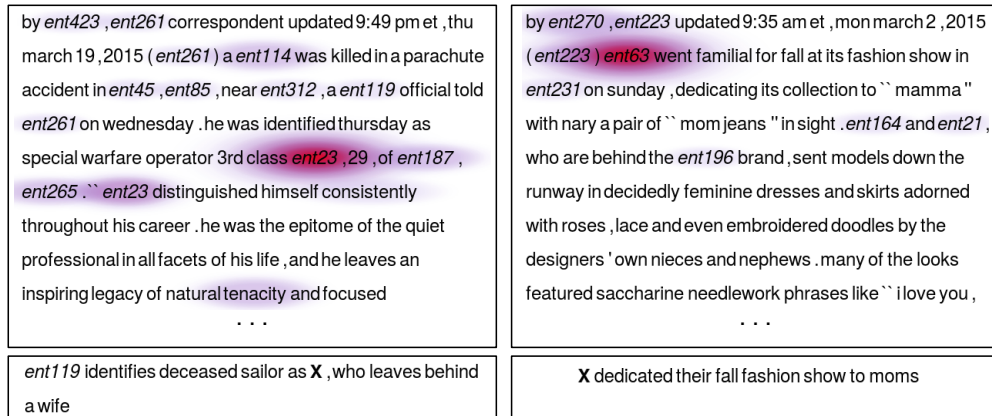

Figure 3: Attention heat maps from the Attentive Reader for two correctly answered validation set queries (the correct answers are *ent23* and *ent63*, respectively). Both examples require significant lexical generalisation and co-reference resolution in order to be answered correctly by a given model.

token integrate long range contextual information via the bidirectional LSTM encoders. Figure 3 depicts heat maps for two queries that were correctly answered by the Attentive Reader.[7] In both cases confidently arriving at the correct answer requires the model to perform both significant lexical generalsiation, e.g. 'killed' → 'deceased', and co-reference or anaphora resolution, e.g. '*ent119* was killed' → 'he was identified.' However it is also clear that the model is able to integrate these signals with rough heuristic indicators such as the proximity of query words to the candidate answer.

## 5   Conclusion

The supervised paradigm for training machine reading and comprehension models provides a promising avenue for making progress on the path to building full natural language understanding systems. We have demonstrated a methodology for obtaining a large number of document-query-answer triples and shown that recurrent and attention based neural networks provide an effective modelling framework for this task. Our analysis indicates that the Attentive and Impatient Readers are able to propagate and integrate semantic information over long distances. In particular we believe that the incorporation of an attention mechanism is the key contributor to these results.

The attention mechanism that we have employed is just one instantiation of a very general idea which can be further exploited. However, the incorporation of world knowledge and multi-document queries will also require the development of attention and embedding mechanisms whose complexity to query does not scale linearly with the data set size. There are still many queries requiring complex inference and long range reference resolution that our models are not yet able to answer. As such our data provides a scalable challenge that should support NLP research into the future. Further, significantly bigger training data sets can be acquired using the techniques we have described, undoubtedly allowing us to train more expressive and accurate models.

## Footnotes

[1] www.cnn.com

[2] www.dailymail.co.uk

[3] http://www.github.com/deepmind/rc-data/

[4]The vocabulary includes all the word types in the documents, questions, the entity maskers, and the question unknown entity marker.

[5]For the Deep LSTM Reader, we consider hidden layer sizes $[64, 128, \underline{256}]$, depths $[1, \underline{2}, 4]$, initial learning rates $[1\text{E}{-}3, 5\text{E}{-}4, \underline{1\text{E}{-}4}, 5\text{E}{-}5]$, batch sizes $[16, \underline{32}]$ and dropout $[0.0, \underline{0.1}, 0.2]$. We evaluate two types of feeds. In the *cqa* setup we feed first the context document and subsequently the question into the encoder, while the *qca* model starts by feeding in the question followed by the context document. We report results on the best model (underlined hyperparameters, *qca* setup). For the attention models we consider hidden layer sizes $[64, 128, 256]$, single layer, initial learning rates $[1\text{E}{-}4, 5\text{E}{-}5, 2.5\text{E}{-}5, 1\text{E}{-}5]$, batch sizes $[8, 16, 32]$ and dropout $[0, 0.1, 0.2, 0.5]$. For all models we used asynchronous RmsProp [20] with a momentum of 0.9 and a decay of 0.95. See Appendix A for more details of the experimental setup.

[6]Memory constraints prevented us from experimenting with deeper Attentive Readers.

[7]Note that these examples were chosen as they were short, the average CNN validation document contained 763 tokens and 27 entities, thus most instances were significantly harder to answer than these examples.

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
