[Supplementary Material]

## A  Model hyperparameters

The precise hyperparameters used for the various attentive models are as in Table 6. All models were trained using asynchronous RmsProp [20] with a momentum of $0.9$ and a decay of $0.95$.

| Model | Hidden Size | Learning Rate | Batch Size | Dropout |
|---|---|---|---|---|
| Uniform, CNN | 256 | 5E-5 | 32 | 0.2 |
| Attentive, CNN | 256 | 5E-5 | 32 | 0.2 |
| Impatient, CNN | 256 | 5E-5 | 32 | 0.3 |
| Uniform, Daily Mail | 256 | 5E-5 | 32 | 0.2 |
| Attentive, Daily Mail | 256 | 2.5E-5 | 32 | 0.1 |
| Impatient, Daily Mail | 256 | 5E-5 | 32 | 0.1 |

Table 6: Model hyperparameters

## B  Performance across document length

To understand how the model performance depends on the size of the context, we plot performance versus document lengths in Figures 4 and 5. The first figure (Fig. 4) plots a sliding window of performance across document length, showing that performance of the attentive models degrades slightly as documents increase in length. The second figure (Fig. 5) shows the cumulative performance with documents up to length $N$, showing that while the length does impact the models' performance, that effect becomes negligible after reaching a length of ~500 tokens.

Figure 4: Precision@Document Length for the attention models on the CNN validation data. The chart shows the precision for each decile in document lengths across the corpus as well as the precision for the 5% longest articles.

Figure 5: Aggregated precision for documents up to a certain lengths. The points mark the $i^{th}$ decile in document lengths across the corpus.

## C  Additional Heatmap Analysis

We expand on the analysis of the attention mechanism presented in the paper by including visualisations for additional queries from the CNN validation dataset below. We consider examples from the Attentive Reader as well as the Impatient Reader in this appendix.

### C.1  Attentive Reader

**Positive Instances**  Figure 6 shows two positive examples from the CNN validation set that require reasonable levels of lexical generalisation and co-reference in order to be answered. The first query in Figure 7 contains strong lexical cues through the quote, but requires identifying the entity

quoted, which is non-trivial in the context document. The final positive example (also in Figure 7) demonstrates the fearlessness of our model.

by *ent362* , *ent300* updated 6:06 pm et , thu march 26 , 2015 ( *ent300* ) the `` *ent321* '' series will have to handcuff a new director . *ent201* , who directed `` *ent71* , '' told *ent286* that she wo n't be back for the sequel , `` *ent100* . '' `` directing ' *ent135* ' has been an intense and incredible journey for which i am hugely grateful , '' she said in a statement to the site . `` while i will not be returning to direct the sequels , i wish nothing but success to whosoever takes on the exciting challenges of films two and three . '' ' *ent71* ' : what fans hoped for ? the first film in the best - selling book series has been hugely successful , pulling in more than $ 550 million worldwide since it premiered in mid-february , but there have been rumbles that creative clashes were in the offing for the sequel . author *ent341* has a great deal of control in how her books are presented on screen , and she made it clear that she wanted to write the screenplay for the second film , *ent184* reported last month . *ent28* wrote the screenplay for `` *ent71* . '' the story behind mr. *ent289* 's suits the film stars *ent344* as billionaire *ent275* -- a man of certain sexual proclivities -- and *ent407* as his romantic partner , *ent389* .

**X** bows out of the `` *ent321* '' sequel

by *ent339* , *ent42* updated 2:59 pm et , thu march 26 , 2015 ( *ent42* ) call it `` *ent351* . '' a *ent396* state trooper caught a driver using a cardboard cutout of *ent421* , the *ent364* beer pitchman known as `` *ent397* . '' the driver , who was by himself , was attempting to use the *ent214* . `` the trooper immediately recognized it was a prop and not a passenger , '' trooper *ent367* told the *ent375* . `` as the trooper approached , the driver was actually laughing . '' *ent143* sent out a tweet with a photo of the cutout -- who was clad in what looked like a knit shirt , a far cry from his usual attire -- and the unnamed laughing driver : `` i do n't always violate the *ent303* lane law ... but when i do , i get a $ 124 ticket ! we 'll give him an a for creativity ! '' the driver was caught on *ent300* near *ent327* , *ent396* , just outside *ent53* . `` he could have picked a less recognizable face to put on his prop , '' *ent143* told the *ent375* . `` we see that a lot . usually it 's a sleeping bag . this was very creative . ''

a driver was caught in the **X** with a cutout of `` *ent7* ''

Figure 6: Attention heat maps from the Attentive Reader for two more correctly answered validation set queries. Both examples require significant lexical generalisation and co-reference resolution to find the correct answers *ent201* and *ent214*, respectively.

**Negative Instances**   Figures 8 and 9 show examples of queries where the Attentive Reader fails to select the correct answer. The two examples in Figure 8 highlight a fairly common phenomenon in the data, namely ambiguous queries, where—at least following the anonymisation process—multiple entities are plausible answers even when evaluated manually. Note that in both cases the query searches for an entity marker that describes a geographic location, preceded by the word "in". Here it is unclear whether the placeholder refers to a part of town, town, region or country.

Figure 9 contains two additional negative cases. The first failure is caused by the co-reference entity selection process. The correct entity, *ent15*, and the predicted one, *ent81*, both refer to the same person, but not being clustered together. Arguably this is a difficult clustering as one entity refers to "Kate Middleton" and the other to "The Duchess of Cambridge". The right example shows a situation in which the model fails as it perhaps gets too little information from the short query and then selects the wrong cue with the term "claims" near the wrongly identified entity *ent1* (correct: *ent74*).

## C.2   Impatient Reader

To give a better intuition for the behaviour of the Impatient Reader, we use a similar visualisation technique as before. However, this time around we highlight the attention at every time step as the model updates its focus while moving through a given query. Figures 10–13 shows how the attention of the Impatient Reader changes and becomes increasingly more accurate as the model

of the officers had to have bullet fragments removed from his arm later , according to the *ent315* . the *ent454* reported that the officers had been driving through the neighborhood dressed in plain clothes . the officers returned fire , and several suspects scattered , *ent195* . *ent47* told the newspaper , adding that the officers believed that they were targeted . but a public information officer for the *ent315* disputes that possibility . `` the officers were in plain clothes ," *ent309* told *ent100* . `` this can not be called targeting . the narcotics officers from the 77th division were driving in an unmarked police vehicle around 64th and *ent223* when they were shot at and they returned fire ." three individuals were detained for questioning , according to *ent309* , but were not arrested . the names of the injured officers have not been released .

**X** _UNK_ : `` this can not be called targeting "

by *ent63* , *ent171* updated 5:59 pm et , tue march 10 , 2015 ( *ent171* ) there was a street named after *ent164* , but they had to change the name because nobody crosses *ent164* and lives . *ent164* counted to infinity . twice . death once had a near - *ent164* experience . *ent164* is celebrating his 75th birthday -- but the calendar is only allowed to turn 39 . that last one is true ( well , the first part , anyway ) . the actor , martial - arts star and world 's favorite tough - guy joke subject was born march 10 , 1940 , which makes him 75 today . or perhaps he is 39 . because maybe you ca n't beat time , but *ent164* can beat anything . happy birthday !

tuesday is **X** ' 75th birthday

Figure 7: Two more correctly answered validation set queries. The left example (entity *ent315*) requires correctly attributing the quote, which does not appear trivial with a number of other candidate entities in the vicinity. The right hand side shows our model is not afraid of Chuck Norris (*ent164*).

by *ent58* , *ent61* updated 11:44 am et , tue march 10 , 2015 ( *ent61* ) a suicide attacker detonated a car bomb near a police vehicle in the capital of southern *ent29* 's *ent85* on tuesday , killing seven people and injuring 23 others , the province 's deputy governor said . the attack happened at about 6 p.m. in the *ent8* area of *ent67* city , said *ent30* , deputy governor of *ent85* . several children were among the wounded , and the majority of casualties were civilians , *ent30* said . details about the attacker 's identity and motive were n't immediately available .

car bomb detonated near police vehicle in **X** , deputy governor says

by *ent18* , for *ent65* updated 7:28 pm et , sat march 28 , 2015 *ent73* , *ent64* ( *ent65* ) suspected *ent53* gunmen decapitated 23 people in a raid on *ent80* village in northeast *ent64* 's *ent24* , residents and a politician said saturday . scores of attackers invaded the village at 11 p.m. friday when residents were mostly asleep and set homes on fire , hacking residents who tried to flee . `` the gunmen slaughtered their 23 victims like rams and decapitated them . they injured several people ," said *ent47* , a local politician who fled .

· · ·

suspected militants raid village in **X**

Figure 8: Attention heat maps from the Attentive Reader for two wrongly answered validation set queries. In the left case the model returns *ent85* (correct: *ent67*), in the right example it gives *ent24* (correct: *ent64*). In both cases the query is unanswerable due to its ambiguous nature and the model selects a plausible answer.

considers larger parts of the query. Note how the attention is distributed fairly arbitraty at first, slowly focussing on the correct entity *ent5* only once the question has sufficiently been parsed.

by *ent25* , *ent63* updated 8:47 pm et , fri march 27 , 2015 ( *ent63* ) enjoy the latest pictures of the former *ent15* . they 're the last you 'll see for a while . *ent36* of *ent31* made her last official appearance friday at a variety of spots across *ent69* , enjoying tours of a learning center and a church that hosts a youth charity . the former , the *ent8* , is named for an aspiring architect who was stabbed to death at age 18 in 1993 . his mother , *ent20* , escorted *ent81* and her husband , prince *ent7* , around the facility . *ent81* , 33 , is scheduled to give birth in mid- to late april , she said this month . it will be the second child for her and *ent7* , 32 . their son , *ent42* , was born in july 2013 .

**X** and *ent7* have a son , *ent42*

by *ent47* , *ent54* and *ent44* , *ent6* updated 8:31 pm et , thu march 26 , 2015 ( *ent6* ) *ent1* has arrested what it claims are two spies who worked for *ent77* 's intelligence service , a *ent70* official said thursday on condition of anonymity . the men , identified as *ent69* and *ent41* , are accused of committing crimes of `` terrorism '' and bringing in `` large quantities of forged currency , '' the *ent70* source said . the official said *ent69* had made a declaration of guilt . *ent6* can not confirm the authenticity of the declaration or whether , if *ent69* made one , it was made under duress . *ent77* 's *ent74* told *ent6* that `` the information you 've obtained is not true . '' `` we do n't have any information that members of nis were arrested in *ent1* , '' an *ent74* representative said .

*ent77* 's **X** denies claim

Figure 9: Additional heat maps for negative results. Here the left query selected *ent81* instead of *ent15* and the right query *ent1* instead of *ent74*.

Figure 10: Attention of the Impatient Reader at time steps 1, 2 and 3.

Figure 11: Attention of the Impatient Reader at time steps 4, 5 and 6.

by *ent20* , *ent48* correspondent updated 9:49 pm et , thu march 19 , 2015 ( *ent48* ) a *ent69* was killed in a parachute accident in *ent31* , *ent52* , near *ent49* , a *ent77* official told *ent48* on wednesday . he was identified thursday as special warfare operator 3rd class *ent5* , 29 , of *ent55* , *ent34* . `` *ent5* distinguished himself consistently throughout his career . he was the epitome of the quiet professional in all facets of his life , and he leaves an inspiring legacy of natural tenacity and focused commitment for posterity , '' the *ent77* said in a news release . *ent5* joined the seals in september after enlisting in the *ent77* two years earlier . he was married , the *ent77* said . initial indications are the parachute failed to open during a jump as part of a training exercise . *ent5* was part of a *ent67* - based *ent69* team .

*ent77* identifies deceased sailor as **X** , who leaves behind a wife

Figure 12: Attention of the Impatient Reader at time steps 7, 8 and 9.

by *ent20* , *ent48* correspondent updated 9:49 pm et , thu march 19 , 2015 ( *ent48* ) a *ent69* was killed in a parachute accident in *ent31* , *ent52* , near *ent49* , a *ent77* official told *ent48* on wednesday . he was identified thursday as special warfare operator 3rd class *ent5* , 29 , of *ent55* , *ent34* . `` *ent5* distinguished himself consistently throughout his career . he was the epitome of the quiet professional in all facets of his life , and he leaves an inspiring legacy of natural tenacity and focused commitment for posterity , '' the *ent77* said in a news release . *ent5* joined the seals in september after enlisting in the *ent77* two years earlier . he was married , the *ent77* said . initial indications are the parachute failed to open during a jump as part of a training exercise . *ent5* was part of a *ent67* - based *ent69* team .

*ent77* identifies deceased sailor as **X** , who leaves behind a wife

Figure 13: Attention of the Impatient Reader at time steps 10, 11 and 12.