[Reviews · NeurIPS 2015]

Submitted by Assigned_Reviewer_1

This paper investigates the challenging problem of reading a text and answering questions about it.

They develop a new large-scale dataset for this machine reading task, designing the dataset so as to focus on the core comprehension task and avoid simplistic solutions.

Experiments with models including simplistic statistical baselines, a basic symbolic-NLP pipeline, and three neural network models confirm that the dataset does require sophisticated solutions to perform well, and that the size is sufficient for NN models to perform well (better than the symbolic-NLP baseline).

The main new result is that attension-based mechanisms are required for NN models to perform well on machine reading.

This is a substantial, well done piece of work.

The authors do a good job of addressing the linguistic issues in designing a dataset that really evaluates the ability to answer factoid-type questions about a text, while still producing a dataset that is large enough for supervised learning models to be competitive.

The neural network models which they propose for this task are clearly based on previous work, but, to my knowledge, are nonethesess novel contributions for this task.

The experiments provide some useful baseline results, and indicate that LSTM models are not sufficient for this task unless they are augmented with an (unsupervised) attension mechanism that idenfifies which parts of the text are relevant to the question.

In the best model, each position in the question has its own attension distribution over the text.

The discussion and the writing are also good.

This paper introduces a data resource that will be valuable to future research in neural network models of text comprehension.

My only real concern is whether NIPS is the right place to publish a paper like this.

It is clearly relevant to current work being done within the ML/NIPS community, but it is less clear that the NIPS reviewing process is the best way to evaluate its contribution.

Summary: This a good, thorough piece of work on machine reading, with important empirical results, some new neural network models, and a new dataset release.

It is certainly relevant to current work being published in the NIPS community, although I am less confident that it is appropriate as "a NIPS paper".

Submitted by Assigned_Reviewer_2

This paper deals with the formal question of machine reading. It proposes a novel methodology for automatic dataset building for machine reading model evaluation. To do so, the authors leverage on news resources that are equipped with a summary to generate a large number of questions about articles by replacing the named entities of it. Furthermore a attention enhanced LSTM inspired reading model is proposed and evaluated. The paper is well-written and clear, the originality seems to lie on two aspects. First, an original methodology of question answering dataset creation, where context-query-answer triples are automatically extracted from news feeds. Such proposition can be considered as important because it opens the way for large model learning and evaluation. The second contribution is the addition of an attention mechanism to an LSTM reading model. the empirical results seem to show relevant improvement with respect to an up-to-date list of machine reading models.
Summary: Well-written paper that addresses a current problem of statistical NLP. The extension of the LSTM model for question answering using an attention mechanism is original and the dataset creation methodology is novel and provide quite convincing experimental performance.

Submitted by Assigned_Reviewer_3

Given the lack of an appropriate dataset, the author provides a new dataset which scraped CNN and Daily Mail, using both the full text and abstract summaries/bullet points. The dataset was then anonymised (i.e. entity names removed). Next the author presents a two novel Deep long-short term memory models which perform well on the Cloze query task.

The quality of the paper is good; however there are some places for improvement. The abstract is too vague and should describe in more detail the new method and dataset. Furthermore the author does not test the models using any well established existing dataset. Another possible point of improvement would be to compare their method against tree based LSTM approaches, which are very similar in structure. Given the overall complexity of the cell structure, a figure to explain the intuition behind the new model should be added. While the author mentions real life environments, there is no test with text that includes entity names. I realize that this is perhaps too much to ask since the authors are trying to tease apart 'understanding text' from 'knowing the background facts', but it would have been ideal (and yes, a *lot* more work) to have both; people are starting to build in external knowledge bases.

It seems like quibbling but I wish I knew more about when and why this method works or doesn't, i.e., some sort of error analysis. I don't find the heat maps particularly helpful, I'd rather know more about how much context is needed, and such implementation details.

The models help to provide a new innovation in Deep LSTM, but the new datasets may ultimately be a more important and original contribution.

Summary: Overall the paper brings an interesting new dataset and model to the field of reading comprehension. There are some points of improvement and clarification which would improve the paper.

Overall, I like the paper in spite if my complaints.

Submitted by Assigned_Reviewer_4

This paper focuses on the reading comprehension task. It proposes to construct a large dataset using news stories and highlights from CNN and Daily Mail. Reading comprehension is formulated as a Cloze test, where a named entity in the news highlight is replaced by a placeholder. The system is expected to predict the missing entity (answer) based on the entire news document (document) and highlight (query).

The paper explores several approaches to address the problem. The baselines are based on entity frequency, frame semantic parsing, and word distance. A set of neural models, including a multi-layer LSTM and LSTMs with attention mechanism, are compared against the baselines. Results show that adding the attention mechanism helps, and the 'impatient reader' model yields the best performance.

The paper is well structured in general. The proposed approach for constructing supervised reading comprehension dataset using news documents and highlights seems interesting. I have a few concerns with the paper, they are listed below.

It seems the baselines used in the paper are pretty weak. The frame semantic parsing and word distance benchmarks are both unsupervised and they seem to be ad-hoc. Although the frame semantic parser is used to retrieve predicate/argument triples, only heuristic rules are used to combine the triples for answer prediction. I feel the state-of-the-art NLP approaches could do much better than that. Perhaps even a supervised extension would generate better results.

I'd also be curious to know the difficulty of the proposed Cloze test. In particular, how well human subjects would perform on the anonymized test. I suspect entities in the news highlight occur frequently in the document. It would be helpful to provide some coverage statistics. For example, the percentage of gold-standard answers that are covered in the top-3 or top-5 most frequent entities. If the coverage is high, perhaps the task is not as difficult as it may seem.

Albeit the good performance, I'm not fully convinced that LSTM would be the method-of-choice

for this task. It seems adding the attention mechanism certainly works, since it helps identify the passages that are related to the answer. However, I'm a bit surprised to see that without the attention mechanism, the single-layer LSTM model yields much worse performance ('uniform attention' in Table 5). I'm also not sure why 'impatient reader' performs well. Some explanations on the advantage of this model would be helpful.

The question categorization in Section 2.1 seems unclear. The definition of different categories needs clarification. Providing example queries may help. If space is a concern, it seems okay to completely remove this section.
Summary: The proposed approach for constructing supervised reading comprehension dataset using news documents and highlights seems interesting. However, I'm concerned about the weak baselines, and because of that, I'm not fully convinced that LSTM is a good mechanism for dealing with such tasks.

Author Feedback
Author rebuttal: We would like to thank all reviewers for their insightful and supportive reviews of our work.

Concerning your comments about the baselines, benchmarks, and additional models that could be applied to this data, we fully agree with these suggestions. By releasing the dataset and the general methodology for creating further question-answering datasets we hope to encourage the wider community to try out a wide range of models on this task. There is certainly more work to be done in building stronger baselines (and models!), both from a NLP and a purely neural side, and we intend to continue contributing to this effort in future work.

While there is a limit to the number of models and baselines we can explore in a single paper, reviewer #3 asked a good question concerning the difficulty of the Cloze form questions, especially in regard with the relative frequency of the entities. We will amend the paper to include more details for this. On both the CNN and Daily Mail dataset, the correct answer is outside of the four most frequent entities more than 50% of the time. However, the concern is a valid point in that the correct answer is contained within the top 10 entities on both datasets 76 percent of the time. While this information doesn't necessarily help in designing an additional baseline, it does help inform on the difficulty of the overall task.

Reviewer #4 raised an interesting issue, pointing out recent work in incorporating external knowledge sources (e.g. Freebase) in question answering. At this point we see our work as complementary to such models, but attempts at combining unstructured context and structured knowledge bases is certainly an interesting avenue for further work!

There were a number of comments concerning clarity of our model presentation as well as the intuition that went into the model design. We will make an effort to address these as well as possible, by better motivating the decisions we made when designing the model (particularly the impatient reader), as well as providing more clarity in the descriptions in the model and experimental sections.

To conclude, we would again like to thank you for your input which should help us make this a better paper. We hope that you will find the updated version of more informative and clearer compared with the original version you reviewed.